# Phonetic Tonal Manifestations and Trends in Tone Change: A Case Study of the Yong-Deng Dialect in Northwest China

Li Yi

College of Humanities and Development Studies, China Agricultural University, Beijing 100083, China; daisy-yili@163.com

**Abstract:** This study takes the Yong-deng dialect as a case study to investigate the phenomenon of tonal merging observed in Northwest Chinese dialects. It begins by examining the various monosyllabic tone patterns of the Yong-deng dialect, then supplements this with a review of the relevant literature, comparisons with the tone patterns of the neighbouring dialects, and an analysis of its tone sandhi in disyllabic and trisyllabic combinations. Each step of the dialect's tonal variation is scrutinised, allowing for the identification of pertinent phonetic biases and the derivation of associated phonological rules. The central argument advanced here is that both synchronic tonal variation and diachronic tone change are governed by specific phonological rules. Despite the ostensibly variable phonetic manifestations, these rules can enable the prediction of the trajectory of tone change. The paper contributes to the understanding of tone merger and highlights its systemic and rule-bound nature.

**Keywords:** tone change; phonetic variants; simplification; insertion

## 1. Introduction

The study of tone change has been conducted from diverse perspectives. For instance, Cao (1998) attributed sound change to both internal and external causes, proposing that any sound change within a language or dialect is due partly to language contact and partly to internal variation. Later, Xu (2001) identified voluntary and involuntary forces as key drivers of tonal variations in connected speech, the former stemming from linguistic/paralinguistic demands, and the latter from articulatory constraints. Hyman (2017) introduced diachronic tone change patterns, such as the spreading of tones perseveratively, the levelling out of contour tones, L-H sequence intervals tending to compress, and H-L sequence intervals tending to expand. Pittayaporn (2018) posits that "diachronic sound changes are result of phonologisation of synchronic patterns of phonetic variation". Zhu et al. (2015) proposed a "tone-evolution clock" indicating the directionality of tone change in Chinese dialects based on a synthesis of existing studies. In an attempt to construct a diachronic typology of tone change, Yang and Xu (2019) undertook a comprehensive review of 52 pertinent studies across 45 language varieties in East and Southeast Asia, with their findings underscoring that the clockwise changes are by far the most common type of tone change.

Yip's (1989) investigation into the application of the Obligatory Contour Principle (OCP) on the sandhi tone in Tianjin dialect suggested that "the first tone changes either by simplification or by insertion". In the context of the sandhi tone, one feature of the initial syllable tends to be either left out or inserted into. Yang and Xu (2019) also proposed "carryover effects" and "truncation" as phonetic mechanisms that trigger tone change, which become observable predominantly across multiple syllables in connected speech. According to Ohala (1989), "sound change is drawn from a pool of synchronic variation". In his approach, sound change is a process comprising three stages: (1) the generation of orderly variation in speech due to phonetic biases in speech production and perception, (2) systematic biases that favour or disfavour certain variants, with favoured variants potentially

becoming phonological rules in an individual's grammar, and (3) the innovative variant being adopted as a norm within the community.

Considering tone change in Northwest Chinese dialects, numerous studies (A. Zhang 2005; S. Zhang 2000; G. Zhang 2012; Zhu and Yi 2015; Yi 2019; Zhai and Zhang 2019) agree on a trend of gradual reduction in citation (or monosyllabic) tone categories. As Zhu and Yi (2015) noted, "the systems of 3- (47%) and 2-tone pattern (6%) constitute a majority in northwest Chinese dialects, which is only one step away from tone language to non-tone language". Many researchers have attributed the tone change in Northwest Chinese dialects predominantly to language contacts (Gao 1980; Liu 2004; Lo 1999; Mo and Zhu 2014). This view seems plausible given the history of Northwest China, in which diverse ethnic communities have settled over the past 2000 years, the majority of whom speak non-tone languages. Nonetheless, Yi (2019) argues that despite external stimuli, the tone merger in Lanzhou dialect still adheres to specific rules, a conclusion drawn from examining the tone-pattern variations of 34 speakers.

Before proceeding, it is essential to clarify the relationship among monosyllabic tone, disyllabic tone and word-tone, the three concepts that have always been involved in the studies of tone change. Chen (2001) defined monosyllabic tone as the tone pattern manifested when a Chinese character is produced in isolation in a monosyllabic form, while disyllabic tone, conversely, refers not only to the tone pattern being present in two syllables but also to the alternation of tonal features in connected speech. Finally, the concept of word-tone, as defined by Xing and Ma (2019), possesses a fixed tonal feature and does not adhere to tone-sandhi rules in connected speech. The fixed word-tone pattern will not be discussed in this article.

Opinions differ among scholars regarding the relationship between monosyllabic and disyllabic tones. Many linguists perceive monosyllabic tone as the underlying representation, with disyllabic tone as its surface counterpart (Cao 1998; Q. Li 2001; X. Li 2002). Chen (2001), however, suggests this relationship is contingent on the specific dialect, arguing that both monosyllabic and disyllabic tones can serve as underlying representations. Some scholars have proposed that disyllabic tones preserve the ancient tonal forms (Mo and Zhu 2014). From a diachronic perspective, Yi (2019) emphasises the significance of tone changes, whereas from a synchronic viewpoint, the alternation between citation tone and tone sandhi is of greater importance.

In this research, I observed a unique case within the Yong-deng dialect, a small county in Gansu province, Northwest China (hereafter abbreviated as YD), where more than one monosyllabic tone-pattern variant exist among different speakers synchronically. This represents a deviation from tone languages. General Lanyin Mandarin dialects typically maintain a stable three-citation-tone or four-citation-tone pattern, whereas YD encompasses seven different tone-pattern variants due to various merging combinations. Yi (2019) reported that the Lanzhou dialect is undergoing changes manifested in two key aspects: (1) Yin-ping (T1a) has two variants, falling and level, with the former being progressively replaced by the latter; (2) Shang-sheng (T2) and Qu-sheng (T3) are merging. Considering that Lanzhou is geographically close to YD, I propose a study on the tonal variations within the YD dialect. I posit that this study will contribute significantly to our understanding of the trends in tone change.

## 2. Materials and Methods

This research was conducted within the YD dialect community. This section discusses the participants, the recording materials and the methods used for data analysis.

### 2.1. Participants

A total of 31 local YD speakers participated in this study, of which 13 females and 18 males are aged from 18 to 79 years old (mean age = 39.77; standard deviation = 17.30). Among them, 8 speakers have graduated from college, 10 have finished high school educa-

tion, 11 have finished secondary education and 2 have received primary education. Further details can be found in Appendices A and B.

*2.2. Materials*

Chen (2001) concluded that most Chinese dialects adhere to a maximal syllable structure of CGVX (a consonant, a glide, a vowel and a coda). These syllables fall into two categories: (1) "checked" syllables ending in an occlusive coda ([-p, t, k]); (2) "smooth" or "slack" syllables characterised with an open syllable CV or a syllable closed by a nasal stop. Based on the division, four tonal categories (Ping 平, Shang 上, Qu 去, Ru 入) were established during the Middle-Chinese period. The four tonal categories have undergone various splits and mergers that are sensitive to different phonological conditions, among which the voicing of onset is the most notable. In each case, the Yang register features a voiced onset and the Yin register features a voiceless onset. In general, Chinese Mandarin dialects have four tonal categories left: T1a (Yin-ping 阴平), T1b (Yang-ping 阳平), T2 (Shang-sheng 上声) and T3 (Qu-sheng 去声). The Ru (入) tone has merged into T1a (Yin-ping 阴平), T1b (Yang-ping 阳平) and T3 (Qu-sheng 去声) depending on different dialects. The investigation manual of Chinese dialects (China Academy of Social Science 2004) was designed in alignment with the tonal categories and the recording materials used for this study were extracted from it. A total of 148 citation words, 292 disyllabic words, and 186 trisyllabic words were selected based on the daily word usage in YD observed in a preliminary investigation.

*2.3. Procedures and Data Analysis*

To avoid the potential effects of "tone sandhi" common in Chinese dialects, this study did not employ the "carrier sentence" methodology proposed by Ladefoged (2003). Sound annotation and parameter extraction were performed using Praat (Boersma and Weenink 2022), with the F0 trajectory of each token manually segmented from vowel onset to offset. The raw F0 values were normalised using Log-Z scores to minimise variation caused by different stature shapes, and feature values were assigned accordingly.

Since T1a and T2 in YD both have two variations, a perception test was adopted. For the test, MATLAB software was used to moderate the pitch contours of T1a and T2, specifically, to alternate the pitch contour of T1a from a high falling to a high level and the pitch contour of T2 from a low dipping to a level. In total, 21 participants (12 females and 9 males) were recruited to take part in this test. Recordings from two speakers (XXF and LLY, each representing one type of T1a and T2) were used as the listening materials.

In an endeavour to investigate whether sociolinguistic factors contribute to phonetic tone variation, a crosstab correlation analysis was conducted examining age, education level, gender, and tone-pattern preference. Preceding this analysis, the variables were assigned values as follows: male = 1, female = 0; age below 50 = 0, age 50 or above = 1; primary education = 0, secondary education=1, high school education = 2 and college education = 3. The tone pattern adopted by more than half the participants was taken as a basic type, and the other tone patterns were taken as variations. I assigned basic type = 1, other variations = 0. Moreover, three chi-square tests were also employed to confirm the extent of correlation.

Drawing from Yip's (1989, 2002) work, in contour tones, the tonal root links to two pitch features, while in level tones, the tonal root connects to one pitch feature. Given the tone patterns of the Northwest Chinese dialects and for the ease of demonstration, this paper adopts a feature representation in tonal description, employing the three features of High (H), Middle (M) and Low (L).

In this article, the symbol "=" is used to indicate that two monosyllabic tonal categories share the same tone value, with the direction of merger specified to demonstrate that the left category merges into the right category.

## 3. Results

*3.1. Monosyllabic, Disyllabic and Trisyllabic Tone Patterns in YD*

3.1.1. Monosyllabic Tone Pattern in YD

Table 1 catalogues one basic monosyllabic tone pattern and six distinct variants identified in YD speakers (refer to Appendix A for more extensive details. Table 1 includes seven representative cases; those sharing a tone pattern have been omitted. The tone pattern of speaker ZMS, which aligns with standard Mandarin, has also been omitted). YD has five different two-tone systems and two three-tone systems. Interestingly, despite sharing the same numerical classification, these systems do not consistently merge tone categories identically. Among these, the basic type is the most prevalent, with 17 speakers subscribing to this tone pattern. The second most common is Variant 1, identified in seven speakers, followed by Variant 2, found in two speakers. For the remaining four variants, each is unique to a single speaker.

**Table 1.** Monosyllabic tone patterns of YD.

| Variant | Tone Numbers | T1a | T1b | T2 | T3 | Speakers |
|---------|--------------|-----|-----|-----|-----|----------|
| Basic Type | 3 | H | HL | =T3 | MLM | 17 |
| Variant 1 | 2 | =1b | HL | =T3 | MLM | 7 |
| Variant 2 | 3 | H | HL | 1a/T3 | MLM | 2 |
| Variant 3 | 2 | H | HL | =1a | =1a | 1 |
| Variant 4 | 2 | =1b | HL | MLM | =1b | 1 |
| Variant 5 | 2 | H | =1a | =1a | HL | 1 |
| Variant 6 | 2 | H | HL | =1a | =1b | 1 |

3.1.2. Disyllabic Tone Pattern in YD

The disyllabic tone patterns of YD are listed in Table 2. Despite the observed variations in monosyllabic tone pattern among different speakers, their disyllabic tone patterns exhibit a commendable degree of consistency.

**Table 2.** Disyllabic tone patterns of YD.

| T Category | T1a | T1b | T2 | T3 |
|------------|-----|-----|-----|-----|
| T1a | H-H | H-HL | H-MLM | H-MLM |
| T1b | HL-HL | HL-HL | HL-MLM | HL-MLM |
| T2 | LM-HL | LM-HL | LM-ML | LM-ML |
| T3 | LM-H | LM-HL | LM-MH | LM-ML |

In the disyllabic tone pattern of YD, T1b remains consistently a high falling [HL]. T1a transitions from a high level [H] to high falling [HL] in the combination of "T1b/T2 +T1a" and preserves the high level in other environments. Both T2 and T3 are low rising [LM] when in the initial positions. In the final positions, they exhibit a low dipping [MLM] when combined with T1a/1b and a low falling [ML] when combined with T2. When combined with T3, T2 transitions to a high rising [MH], while T3 remains a low falling [ML]. Based on these observations, I propose that T1a, T2 and T3 in YD undergo the following phonological processes in disyllabic combinations:

(1)    T1a: H→HL/T__] (T = T1b, T2) (insertion)
(2)    T2/T3: MLM→LM/[__T (T = T1a, T1b, T2, T3) (simplification)
(3)    T2: MLM→ML/T__] (T = T2) (simplification)
(4)    T2: MLM→LM→MH/T__] (T = T3) (simplification + raising)
(5)    T3: MLM→ML/T__] (T = T2, T3) (simplification)

3.1.3. Trisyllabic Tone Pattern in YD

The trisyllabic tonal combinations of YD are listed in Table 3. The trisyllabic combinations, similar to the disyllabic tone pattern in the dialect, demonstrate a high degree of consistency across different speakers.

**Table 3.** Trisyllabic tonal combinations of YD.

| T Category | Feature | T Category | Feature |
|---|---|---|---|
| 1a + 1a + 1a | H-H-H | T2 + 1a + 1a | LM-H-H |
| 1a + 1a + 1b | H-H-HL | T2 + 1a + 1b | LM-H-HL |
| 1a + 1a + T2 | H-H-MLM | T2 + 1a + T2 | ML-H-LM |
| 1a + 1a + T3 | H-H-MLM | T2 + 1a + T3 | LM-ML-LM |
| 1a + 1b + 1a | H-HL-H | T2 + 1b + 1a | LM-HL-H |
| 1a + 1b + 1b | H-HL-HL | T2 + 1b + 1b | LM-HL-HL |
| 1a + 1b + T2 | H-HL-MLM | T2 + 1b + T2 | LM-HL-LM |
| 1a + 1b + T3 | H-HL-MLM | T2 + 1b + T3 | LM-HL-LM |
| 1a + T2 + 1a | H-LM-H | T2 + T2 + 1a | LM-LM-H |
| 1a + T2 + 1b | H-ML-HL | T2 + T2 + 1b | ML-LM-HL |
| 1a + T2 + T2 | H-LM-LM | T2 + T2 + T2 | LM-LM-LM |
| 1a + T2 + T3 | H-LM-MLM | T2 + T2 + T3 | LM-LM-LM(ML) |
| 1a + T3 + 1a | H-ML-H | T2 + T3 + 1a | LM-MLM-H |
| 1a + T3 + 1b | H-ML-HL | T2 + T3 + 1b | LM-MLM-HL(H) |
| 1a + T3 + T2 | H-ML-LM | T2 + T3 + T2 | LM-MLM-LM |
| 1a + T3 + T3 | H-LM-MLM | T2 + T3 + T3 | LM-ML-LM |
| 1b + 1a + 1a | HL-H-H | T3 + 1a + 1a | ML(M)-H-H |
| 1b + 1a + 1b | ML-H-HL | T3 + 1a + 1b | ML-H-HL |
| 1b + 1a + T2 | HL(ML)-H-LM | T3 + 1a + T2 | ML-H-LM |
| 1b + 1a + T3 | HL(ML)-H-LM | T3 + 1a + T3 | ML-H-LM |
| 1b + 1b + 1a | HL-ML-H | T3 + 1b + 1a | ML-HL-H |
| 1b + 1b + 1b | ML-ML-HL | T3 + 1b + 1b | ML-HL-HL |
| 1b + 1b + T2 | ML-HL-LM | T3 + 1b + T2 | ML-HL-LM |
| 1b + 1b + T3 | ML-HL-LM | T3 + 1b + T3 | ML-HL-LM |
| 1b + T2 + 1a | HL-LM-H | T3 + T2 + 1a | ML-LM-H |
| 1b + T2 + 1b | ML-LM-HL | T3 + T2 + 1b | ML-LM-HL |
| 1b + T2 + T2 | HL-ML-LM | T3 + T2 + T2 | ML-ML-LM |
| 1b + T2 + T3 | HL(ML)-LM-MLM | T3 + T2 + T3 | ML-LM-MLM |
| 1b + T3 + 1a | HL-MLM-H | T3 + T3 + 1a | LM-MLM-H |
| 1b + T3 + 1b | HL-ML-HL | T3 + T3 + 1b | LM-MLM-HL |
| 1b + T3 + T2 | HL(ML)-ML-LM | T3 + T3 + T2 | LM-MLM-LM |
| 1b + T3 + T3 | HL-ML-LM | T3 + T3 + T3 | LM-MLM-MLM |

T1a shifts from a high level [H] to a low falling [ML] in the "T2 + T1a + T3" combination, but retains a high level [H] in other contexts. T1b adopts a low falling pattern when situated in the middle position of the "T1b + T1b + T1a/1b" combination and maintains a high falling [HL] pattern in other circumstances. T2 and T3 exhibit changes into either a low rising [LM] or a low falling [ML], depending on their specific combinations. Accordingly, I propose

that T1a, T1b, T2 and T3 in YD undergo the following phonological processes in trisyllabic tone combinations (where $T^I$ represents the initial tone, $T^M$ the middle tone and $T^F$ the final tone):

(6)  T1a: H→ML/$T^I$__$T^F$ ($T^I$ = T2; $T^F$ = T3) (insertion and lowering)

(7)  T1b: HL→ML/$T^I$ __$T^F$ ($T^I$ = 1b; $T^F$ = 1a/1b) (lowering)

(8)  T2: MLM→LM/[__$T^M$+$T^F$ ($T^M$ = 1a; $T^F$ = 1a/1b/T3); ($T^M$ = 1b/T3; $T^F$ = ANYTONE); ($T^M$ = T2; $T^F$ = 1a/T2/T3) (simplification)

(9)  T2: MLM→ML/[__$T^M$+$T^F$ ($T^M$ = 1a; $T^F$ = T2); ($T^M$ = T2; $T^F$ = 1b) (simplification)

(10)  T2: MLM→ML/$T^I$ __$T^F$ ($T^I$ = 1a; $T^F$ = 1a/T3); ($T^I$ = 1b; $T^F$ = T2); ($T^I$ = T3; $T^F$ = T2) (simplification)

(11)  T2: MLM→LM/$T^I$ __$T^F$ ($T^I$ = 1a; $T^F$ = 1b/T2); ($T^I$ = 1b/T3; $T^F$ = 1a/1b/T3); ($T^I$ = T2; $T^F$ = ANYTONE) (simplification)

(12)  T2: MLM→LM/$T^I$+$T^M$__] ($T^I$ = 1a; $T^M$ = T2/T3); ($T^I$ = 1b/T2/T3; $T^M$ = ANYTONE) (simplification)

(13)  T3: MLM→ML/[__$T^M$+$T^F$ ($T^M$ = 1a/1b/T2; $T^F$ = ANYTONE) (simplification)

(14)  T3: MLM→LM/[__$T^M$+$T^F$ ($T^M$ = T3; $T^F$ = ANYTONE) (simplification)

(15)  T3: MLM→ML/$T^I$__$T^F$ ($T^I$ = 1a; $T^F$ = 1a/1b/T2); ($T^I$ = 1b; $T^F$ = 1b/T2/T3); ($T^I$ = T2; $T^F$ = T3) (simplification)

(16)  T3: MLM→LM/$T^I$ __$T^F$ ($T^I$ = 1a; $T^F$ = T3) (simplification)

(17)  T3: MLM→LM/$T^I$+$T^M$__] ($T^I$ = 1b; $T^M$ = 1a/1b/T3); ($T^I$ = T2; $T^M$ = ANYTONE); ($T^I$ = T3; $T^M$ = 1a/1b) (simplification)

### 3.2. Perception Test and Correlation Analysis

3.2.1. Perception Test on T1a and T2

In the YD dialect, T1b is the most stable tone category, consistently exhibiting a high falling across different speakers and in disyllabic and trisyllabic combinations. T3 is less stable; in the monosyllabic tone pattern among different speakers, it often presents as a low dipping tone, while in disyllabic and trisyllabic combinations, it experiences alternations. T1a and T2 display greater variability. Each exhibits two phonetic variations in the monosyllabic tone pattern among different speakers: T1a alternates between a high level and a high falling, while T2 shifts between a high level and a low dipping.

As mentioned above, to explore this further, a perception test was designed. For each of the 16 T1a characters, the participants were presented with two different tone contours, a high level and a high falling. If they believed the high level was correct, they would choose "a"; if they thought the high falling was correct, they would choose "b". If they found both sounds acceptable, they would choose "c", and if neither option was to their preference, they would choose "d". Altogether, the 21 participants made 336 choices. Of these, "a" was selected 167 times, "b" was selected 121 times, "c" was selected 27 times, and "d" was selected 21 times.

For the 18 T2 characters, the participants made choices following the same pattern. Out of the 378 choices, 128 selected the high level as the characteristic feature of T2 and 199 chose the low dipping. "Both" was chosen 32 times and "neither" was selected 19 times (as shown in Figure 1).

These results indicate that the high-level variation of T1a and the low dipping variation of T2 are more widely accepted. However, each only accounts for 50% of the situations, suggesting that the other variation also plays a significant role.

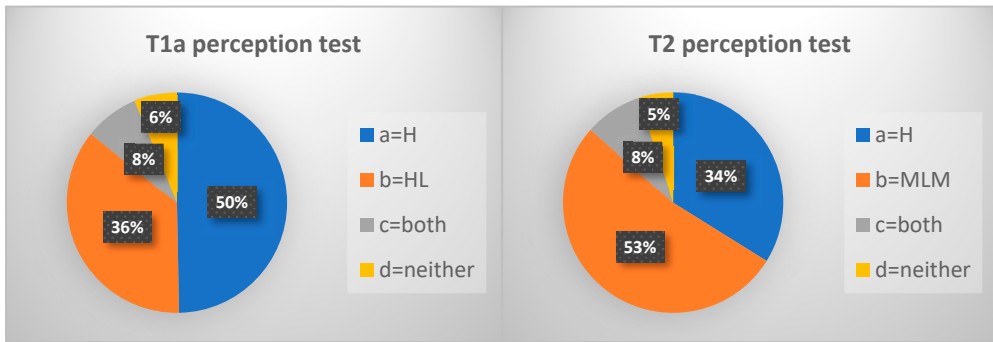

**Figure 1.** The results of perception tests.

3.2.2. Correlation Analysis

The results of the crosstab analysis, as shown in Table 4, reveal the relationship between each qualitative variable and tone-pattern choice. Participants under the age of 50 years old exhibit a bias toward the basic tone pattern, whereas those aged 50 or above are more inclined to adopt other variations. While education level does not appear to exhibit a pronounced connection with tone-pattern preferences, the frequency count indicates that individuals with a high school education lean more toward selecting the basic type. Notably, female speakers are more prone to the basic type, while male speakers demonstrate a predilection for other variations.

**Table 4.** Crosstab correlation of age, education, gender and tone pattern.

| | | | Gender | | Education | | | | Age | | Total |
|---|---|---|---|---|---|---|---|---|---|---|---|
| | | | 0 | 1 | 0 | 1 | 2 | 3 | 0 | 1 | |
| Tone variants | 0 | Count | 3 | 11 | 1 | 7 | 1 | 5 | 6 | 8 | 14 |
| | | % | 9.7% | 35.5% | 3.2% | 22.6% | 3.2% | 16.1% | 19.4% | 25.8% | 45.2% |
| | 1 | Count | 10 | 7 | 1 | 4 | 9 | 3 | 14 | 3 | 17 |
| | | % | 32.3% | 22.6% | 3.2% | 12.9% | 29.0% | 9.7% | 45.2% | 9.7% | 54.8% |

The results of the chi-square tests are shown in Table 5. In terms of the correlation between gender and tone-pattern choice, the chi-squared value is 4.409, with a *p*-value of 0.036, signifying a statistically significant correlation at the 5% level. A similar conclusion is drawn for the correlation between age and tone-pattern choice, yielding a chi-square value of 5.231 and a *p*-value of 0.022. In the context of the association between level of education and tone-pattern choices, the chi-square test value is 7.498, with a *p*-value of 0.058, indicating an absence of significant correlation between them ($p > 0.05$).

**Table 5.** Chi-square tests of age, education, gender and tone-pattern choices.

| | Gender | | Education | | Age | |
|---|---|---|---|---|---|---|
| | Value | Asymp. Sig. (2-sided) | | | | |
| Pearson Chi-Square | 4.409 | 0.036 | 7.498 | 0.058 | 5.231 | 0.022 |
| N of Valid Cases | 31 | | 31 | | 31 | |

*3.3. Prototypes of Monosyllabic Tone Categories in YD*

To ensure a comprehensive understanding of the directionality of mergers in every variation, it is critical to first establish the original prototype of each tone category within the dialect. To this end, three aspects are proposed for comprehensive evaluation: (1) examination of documentary records; (2) comparison with the tone patterns of neighbouring dialects; (3) evaluation of the correspondence among monosyllabic, disyllabic and trisyllabic tone patterns.

Reviewing documentary records and comparing tone patterns from neighbouring dialects were designed to trace diachronic tonal development, while the third approach aimed to discern synchronic tone feature alternations.

Table 6 below presents the tone patterns of dialects neighbouring the YD dialect under study (Yi 2019). Documentary records of YD are provided at the base of each category in numerical representation (Huang and Zhao 1960).

**Table 6.** Tone patterns of neighboring dialects and documentary records of YD.

| County | Feature | | | |
|---|---|---|---|---|
| | **T1a** | **T1b** | **T2** | **T3** |
| Gaolan | HM | HL | LML | MLM |
| Honggu | H | HL | MLM | MLM |
| Lanzhou | H | HL | MLM | MLM |
| YD | 53 | 52 | 442 | 213 |

In the process of establishing the prototype for each tone category, it is vital to consider four key points: (1) stability across all sources; (2) judgement of the underlying representation in synchronic feature alternations, specifically, its prevalence across most environments; (3) the patterns adopted by older speakers, according to literary records and correlation tests; (4) the patterns favoured by the majority of participants in perception tests.

Tone T1b remains a high falling [HL] across all sources, with minor alternation observed in trisyllabic combinations. T3 is relatively consistent. It is classified as low dipping in documentary records and neighbouring dialects, yet experiences phonological changes in disyllabic and trisyllabic combinations. Out of the 56 positions analysed, along with disyllabic and trisyllabic combinations, there are 18 low rising [LM], 20 low falling [ML], and 18 low dipping [MLM] occurrences. As there is no significant difference among these three alternations, the decision was made to follow other sources (monosyllabic tone feature, documentary records and neighbouring dialect references) and assign low dipping as its prototype.

In YD, Tone T1a exhibits two feature values in documentary records and neighbouring dialects: a high-mid falling [HM] and a high level [H]. These same values are also found in disyllabic and trisyllabic combinations. Out of 56 positions, a high falling tone is observed twice and a low falling tone once, with the rest all high level. Moreover, in the perception test, 50% of participants selected a high level tone instead of a high falling tone (as shown in Figure 1). Thus, a high level tone is selected as its prototype.

Tone T2 in YD presents the most uncertainty. Documentary records and neighbouring dialects describe it in three different ways: (1) as a low dipping tone, (2) as a mid-level tone [M] and (3) as a convex tone [LML]. The monosyllabic T2 exhibits two variations: (1) a high level and (2) a low dipping. In the perception test, 53% of participants chose a low dipping tone and 34% chose a high level. In disyllabic and trisyllabic combinations, it appears as a low rising tone 46 times, a low falling tone 6 times and a low dipping tone 4 times. Correlation tests reveal that younger participants tend to adopt the basic type, while older individuals tend to adopt the other six variants, where T2 more often presents as a high level tone. A process of elimination was employed, first, to exclude the least frequent occurrences—convex, low dipping and low falling—then, the relatively recent representation of a low rising tone was excluded, leaving only the level tone.

Combined with all the points discussed above, the prototypical tone features are deduced as T1a = H, T1b = HL, T2 = M and T3 = MLM for the YD dialect (refer to Table 7). As the variations of T2 are all categorised as low tones, a middle level value [M] is temporarily assigned instead of a high level.

**Table 7.** The prototype of monosyllabic tones of HY and YD.

| Areas | T1a | T1b | T2 | T3 |
|---|---|---|---|---|
| YD | H | HL | M | MLM |

*3.4. Tonal Variations in YD*

There are a total of seven different types of tone mergers observed in YD, two of which involve the merger between two tone categories, while the remaining five involve mergers among three tone categories. Each of these types was examined using the same process, starting with the mergers between two categories:

| Basic type | T1a = H | T1b = HL | T2 = T3 | T3 = MLM |
|---|---|---|---|---|
| Variant 2 | T1a = H | T1b = HL | T2 = T1a | T3 = MLM |
| SDW | T1a = H | T1b = HL | T2 = T1a/T3 | T3 = MLM |

In the basic type, T2 and T3 exhibit the same tonal feature, while in Variant 2, T2 and T1a share a common feature. However, there is one speaker (SDW) whose T2 characters display a combination of both features, with half resembling T3 and the other half resembling T1a. To analyse this further, Table 8 shows the two potential merger directions: (1) T2 merging into T3 (as seen in the basic tone pattern) or (2) T1a (as seen in Variant 2), or potentially merging into both (as in the case of SDW). Based on this analysis, I propose that T2 in YD underwent the following changes:

(18)　T2: M→ML→MLM (insertion)
(19)　T2: M→H (raising)

**Table 8.** Tone merger assumption of 3-tone category (note: * refers to the wrong assumption).

| basic type | merger assumption I | | | | merger assumption II | | | |
|---|---|---|---|---|---|---|---|---|
| | T1a | T1b | **T2** | **T3 = T2** | T1a | T1b | **T2 = T3** | T3 |
| | H/HM | HL | **M** | ***M** | H | HL | **MLM** | MLM |
| Variant 2 | merger assumption I | | | | Mmerger assumption II | | | |
| | **T1a = T2** | T1b | **T2** | T3 | T1a | T1b | **T2 = 1a** | T3 |
| | ***M** | HL | **M** | MLM | H | HL | **H** | MLM |
| SDW | merger assumption I | | | | merger assumption II | | | |
| | T1a | T1b | **T2 = 1a** | T3 | T1a | T1b | **T2 = T3** | T3 |
| | H/HM | HL | **H** | MLM | H | HL | **MLM** | MLM |

These changes reflect the evolution of T2 in YD, involving a progression from a mid-level tone to a sequence of mid falling to low dipping (insertion), and alternatively, a shift from mid level to high level (raising).

Now let us examine the tone-pattern variants resulting from three-category mergers in YD:

| Variant 1 | 1a = 1b | 1b = HL | T2 = T3 | T3 = MLM |
|---|---|---|---|---|
| Variant 3 | 1a = H | 1b = HL | T2 = 1a | T3 = 1a |
| Variant 4 | 1a = 1b | 1b = HL | T2 = MLM | T3 = 1b |
| Variant 5 | 1a = H | 1b = 1a | T2 = 1a | T3 = HL |
| Variant 6 | 1a = H | 1b = HL | T2 = 1a | T3 = 1b |

Similar to the previous analysis, I present two assumptions regarding the mergers (see Table 9), with the correct one aligning with the actual survey results. T2 in Variant 3, Variant 5 and Variant 6 underwent the same change as illustrated in (19), while T2 in Variant 1 and Variant 4 experienced the same change as described in (18). T3 in Variant 4, Variant 5 and Variant 6 merged into T1b and underwent the following change:

(20)　T3: MLM→ML→HL (simplification + raising)

T3 in Variant 3 merged into T1a and underwent the following change:

(21)　T3: MLM→M→H (simplification + raising)

T1a in Variant 1 and Variant 4 merged into T1b and underwent the following change:

(22)  T1a:H→HL (insertion)
        T1b in Variant 5 merged into T1a and underwent the following change:
(23)  T1b: HL→H (simplification)

**Table 9.** Tone merger assumption of 2-tone category. (note: * refers to the wrong assumption).

| | merger assumption I | | | | merger assumption II | | | |
|---|---|---|---|---|---|---|---|---|
| **Variant 6** | **1a = T2** | **1b = T3** | T2 | T3 | 1a | 1b | **T2 = 1a** | **T3 = 1b** |
| | ***M** | ***MLM** | M | MLM | H | HL | **H** | **HL** |
| | merger assumption I | | | | merger assumption II | | | |
| **Variant 1** | 1a | **1b = 1a** | T2 | **T3 = T2** | **1a = 1b** | 1b | **T2 = T3** | T3 |
| | H/HM | ***H** | M | ***M** | **HL** | HL | **MLM** | MLM |
| | merger assumption I | | | | merger assumption II | | | |
| **Variant 3** | **1a = T2/T3** | 1b | T2 | T3 | 1a | 1b | **T2 = 1a** | **T3 = 1a** |
| | ***M/MLM** | HL | M | MLM | H | HL | **H** | H |
| | merger assumption I | | | | merger assumption II | | | |
| **Variant 4** | 1a | **1b = 1a** | T2 | **T3 = T2** | **1a = 1b** | 1b | **T2 = T3** | **T3 = 1b** |
| | H/HM | ***H/HM** | M* | ***M** | **HL** | HL | **MLM** | HL |
| | merger assumption I | | | | merger assumption II | | | |
| **Variant 5** | **1a = 1b/T2** | **1b = T3** | T2 | T3 | 1a | **1b = 1a** | **T2 = 1a** | **T3 = 1b** |
| | ***HL/M** | ***MLM** | M | MLM | H | H | H | HL |

The cases of Variant 4 and Variant 5 are slightly more intricate. In Variant 4, T1a and T3 merged into T1b, resulting in a high falling tone, while T2 underwent the same merger process as depicted in (18), resulting in a dipping tone. In the case of Variant 5, T3 merged into T1b (similar to Variant 4 and Variant 6), while T1b merged into T1a, resulting in a high level tone. Notably, this is the only instance where T1b merged into another tone category.

## 4. Discussion

The review conducted by Yang and Xu (2019) highlights prominent cross-linguistic trends in tone change. Specifically, clockwise changes are by far the most prevalent type. The directionality of the clockwise cycle is as follows:

> *low level 11|22 > low falling 32 > mid falling 42 > high falling 52 > high level 55 or rising-falling 453 > mid rising 45|35 > low rising 24|13 > falling-rising 323|214 or low level 11|22 > low falling 32* (Yang and Xu 2019)

An alternative perspective on tone change is presented by Zhu (2018) in his "tone-evolution clock" (Figure 2), which demonstrates a more complex directionality of tone change. For instance, the shift from a high falling tone to a high level tone aligns with the "clockwise" pattern proposed by Yang and Xu (2019). However, within Zhu's tone-evolution clock, these changes are bidirectional, meaning that a high falling tone can transition into a high level tone and vice versa. Similarly, the change from a dipping tone [MLM] can lead to either a low falling tone [ML] or a low rising tone [LM], with the former adhering to the "clockwise route" and the latter occurring in the opposite direction. While in the clockwise pattern, the sequence "low falling > low rising > low falling-dipping" represents a unidirectional alternation, the "tone-evolution clock" allows for the possibility of bidirectional change. Based on this perspective, I am inclined to favour the "tone-evolution clock" framework.

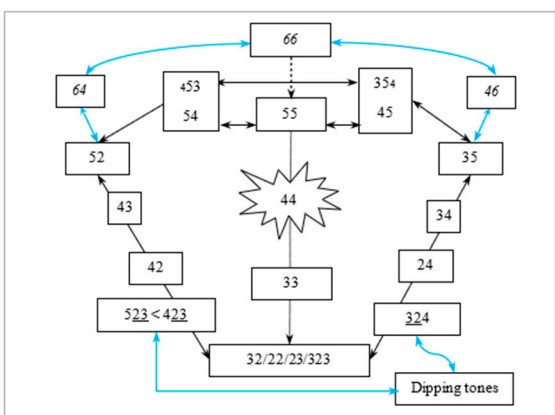

**Figure 2.** The tone-evolution clock (cited in Zhu 2018).

The question arises as to the significance of summarising the directionality of tone change when both patterns, the "clockwise" and the "tone-evolution clock," occur. While the "clockwise" pattern is derived statistically, it fails to elucidate the underlying causes of tone change. Taking T2 in Northwest Chinese dialects as an example, it merges into various tone categories, including T1a, T1b and T3, each exhibiting distinct tone contours. According to Yi (2019), the recorded instances of T2 in Lanyin Mandarin display different positions within the "clockwise circle", ranging from high level to high level-falling, convex and low dipping. Yi (2019) further notes that in the west of Lanzhou, there are five dialect points where T1a and T2 merge, ten dialect points east of Lanzhou and fifteen dialect points west of Lanzhou where T1b and T2 merge (as depicted in Table 10). In the Lanzhou dialect, T2 merges into T3 (Yi 2019). In YD, as shown in Tables 8 and 9, T2 merges into T3 or T1a or both. Interestingly, regardless of the contour of the target tone category, T2 merges into it.

**Table 10.** Three types of T2 merging in Northwest dialects (Cited in Yi 2019).

| T1a | T1b | T2 | T3 | Dialect Points |
|-----|-----|-----|-----|----------------|
| H | MH/HL | =T1a | MLM/HL/LM | 5 LZ west |
| H/HM | MH/HL/MLM | =T1b | HL/LM/H | 10 LZ east; 15 LZ west |
| H/HM | HL | =T3 | MLM | LZ, YD |

(NOTE: "=" indicates " the same as"; LZ = Lanzhou).

These findings suggest that the specific tonal contour of the target category does not determine the merging outcome, as T2 assimilates into various contours within different dialects.

The correlation test (Section 3.2.2) reveals that speakers exhibiting the basic tone-pattern type in YD are generally younger, while older speakers tend to adopt the other tone-pattern variants, with five of them being two-tone patterns. However, according to the literature (Huang and Zhao 1960), YD was a four-tone pattern dialect. Interestingly, I observed 79-year-old and 70-year-old speakers who both exhibit two-tone patterns, as well as an 18-year-old speaker with a two-tone pattern. This leads me to assume that tone mergers have been occurring and are still ongoing. In terms of phonetic representations, T2 manifests as a high level and low dipping tone in monosyllabic patterns and as a low rising and low falling tone in disyllabic and trisyllabic combinations. Similarly, T1a has two phonetic representations: high level and high falling. Upon reviewing the phonetic tonal manifestations in YD, I noticed that the tone changes, both synchronically and diachronically, align with the phonological rules elucidated by Yip (1989). Four rules can be identified: (1) Dipping tones, independent of the tone category, are highly susceptible to alteration in disyllabic and trisyllabic combinations, transitioning either into a low rising

or a low falling tone, both of which adhere to the rule of "simplification". (2) High level tones change into high falling tones, and vice versa, with the former following the rule of "insertion" and the latter following the rule of "simplification". (3) Rules of "raising" and "lowering" occur only as accompaniments. (4) The feature changes observed in tone mergers within monosyllabic tone pattern also appear in disyllabic and trisyllabic combinations.

Yi (2019) found that in the Lanzhou dialect, with the two variations of T1a (high level and high falling), people under 50 years old and people with higher education are more inclined to adopt the level variant. It is within the last 50 years that Standard Mandarin (Putonghua) began to take its prestigious position. In YD, a parallel transformation can also be observed in T1a. Among 31 speakers, 23 opt for the high level variant. And it retains a high level in almost all positions of the disyllabic and trisyllabic combinations. In the perception test, of all the choices, 50% chose high level [H] as the "accepted" tone feature and only 36% chose high falling. It seems that T1a has been influenced by Standard Mandarin, experiencing the alternation from a high-mid falling to a high level. However, the other three tonal categories in YD showed little evidence of the same influence. Moreover, the change from a high falling to a high level also occurs in other dialects/languages and also in other tonal categories (Yang and Xu 2019), where there is not the influence from Chinese Mandarin. I admit that language contact does play a certain role in tone change, while, based on these observations, I propose that the directionality of tone change is influenced more by the phonological rules than by adhering strictly to a "clockwise" or "counter-clockwise" directionality or by the influence of certain language contacts. Furthermore, it is noteworthy that contextual variations in tone sandhi and tone mergers in the monosyllabic tone patterns both follow the same phonological rules.

## 5. Conclusions

In the tone patterns exhibited by YD speakers in this study, there was considerable phonetic variability. However, one aspect remained consistent: the phonetic variations conformed to three phonological rules: (1) simplification, (2) insertion and (3) raising/lowering. Furthermore, the analyses conducted indicate that alternations observed in synchronic tone sandhi also occur in diachronic tone mergers. These interactions between synchronic and diachronic phenomena contribute to the overall trend of tone change. This observation aligns with the "labour-saving principle" proposed by Ohala (1989), which suggests that sound changes emerge from a pool of synchronic variation present in spontaneous speech.

During this process, certain variation biases can be observed. For instance, dipping tones are more likely to transform into low rising or low falling tones, independent of the tone category. Among all the tone categories in YD, T2 stands out as the most active and variable and this observation is supported by Yi's (2019) research, which highlighted the merger of T2 and T3 in the Lanzhou dialect. It can be speculated that T2 in Lanyin Mandarin is also a highly active tone category undergoing a process of tone-category alternation. As this process is still ongoing in the region, further investigation will yield a comprehensive understanding of tone evolution.

**Funding:** This research was funded by the Social Science Foundation of the Ministry of Education of China, grant number 23YJA740049.

**Informed Consent Statement:** Informed consent was obtained from all subjects involved in the study.

**Data Availability Statement:** Acoustic data can be retrieved from https://doi.org/10.5281/zenodo.10 071323 (accessed on 29 January 2023).

**Conflicts of Interest:** The author declares no conflict of interest.

## Appendix A. Tone Values and Other Information of 31 Speakers in YD

| No. | Type | Township/Names | Tone Number | 1a | 1b | T2 | T3 | Gender | Age | Education | Occupation |
|---|---|---|---|---|---|---|---|---|---|---|---|
| 1 | BSC T | Heqiao LWT | 3 | H | HL | MLM | MLM | F | 32 | college | teacher |
| 2 | BSC T | Heqiao THL | 3 | H | HL | MLM | MLM | F | 57 | S school | farmer |
| 3 | BSC T | Heqiao XYY | 3 | H | HL | MLM | MLM | F | 22 | college | student |
| 4 | BSC T | Heqiao YSQ | 3 | H | HL | MLM | MLM | M | 53 | H school | driver |
| 5 | BSC T | Heqiao YYQ | 3 | H | HL | MLM | MLM | F | 41 | S school | farmer |
| 6 | BSC T | Heqiao ZJP | 3 | H | HL | MLM | MLM | M | 18 | H school | student |
| 7 | BSC T | Heqiao ZCG | 3 | H | HL | MLM | MLM | M | 34 | P school | farmer |
| 8 | BSC T | Heqiao HYX | 3 | H | HL | MLM | MLM | F | 36 | college | teacher |
| 9 | BSC T | Heqiao ZYF | 3 | H | HL | MLM | MLM | F | 25 | H school | farmer |
| 10 | BSC T | Liancheng MZJ | 3 | H | HL | MLM | MLM | M | 20 | H school | student |
| 11 | BSC T | Liancheng XYL | 3 | H | HL | MLM | MLM | F | 51 | S school | farmer |
| 12 | BSC T | Liancheng SJH | 3 | H | HL | MLM | MLM | F | 19 | H school | student |
| 13 | BSC T | Liancheng WC | 3 | H | HL | MLM | MLM | M | 19 | H school | student |
| 14 | BSC T | Liancheng WSP | 3 | H | HL | MLM | MLM | M | 19 | H school | student |
| 15 | BSC T | LianchengWYY | 3 | H | HL | MLM | MLM | F | 19 | H school | student |
| 16 | BSC T | Liancheng YFL | 3 | H | HL | MLM | MLM | F | 30 | H school | teacher |
| 17 | BSC T | Chengguan LJC | 3 | H | HL | MLM | MLM | M | 34 | S school | worker |
| 18 | VRT 1 | Liancheng LXD | 2 | H | H/HL | MLM | MLM | M | 34 | college | teacher |
| 19 | VRT 1 | Chengguan FSL | 2 | HL | HL | MLM | MLM | F | 41 | S school | worker |
| 20 | VRT 1 | ChengguanWZX | 2 | HL | HL | MLM | MLM | M | 79 | S school | worker |
| 21 | VRT 1 | Chengguan BZ | 2 | HL | HL | MLM | MLM | M | 61 | S school | worker |
| 22 | VRT 1 | Heqiao LLY | 2 | HL | HL | MLM | MLM | F | 60 | S school | farmer |
| 23 | VRT 1 | Heqiao YTCH | 2 | HL | HL | MLM | MLM | M | 60 | S school | farmer |
| 24 | VRT 1 | Heqiao LTY | 2 | HL | HL | MLM | MLM | M | 48 | college | teacher |
| 25 | VRT 2 | Liancheng ZJH | 3 | H | HL | H | MLM | M | 70 | college | teacher |
| 26 | VRT 2 | Heqiao SDW | 3 | H | HL | H/MLM | MLM | M | 28 | S school | worker |
| 27 | VRT 3 | Liancheng XXF | 2 | H/HL | H/HL | H | H | F | 58 | S school | farmer |
| 28 | VRT 4 | Heqiao YTC | 2 | HL | HL | MLM | HL | M | 51 | P school | farmer |
| 29 | VRT 5 | Heqiao LAX | 2 | H | H | H | HL | M | 18 | college | teacher |
| 30 | VRT 6 | Heqiao ML | 2 | H | HL | H | HL | M | 36 | H school | student |
| 31 | PTH | Heqiao ZMS | 4 | H | MH | MLM/H | HL | M | 32 | college | teacher |

(NOTE: BSC T = basic type; VRT = variant; PTH = Putonghua.)

## Appendix B. The Normalised Tone Patterns of 31 YD Speakers

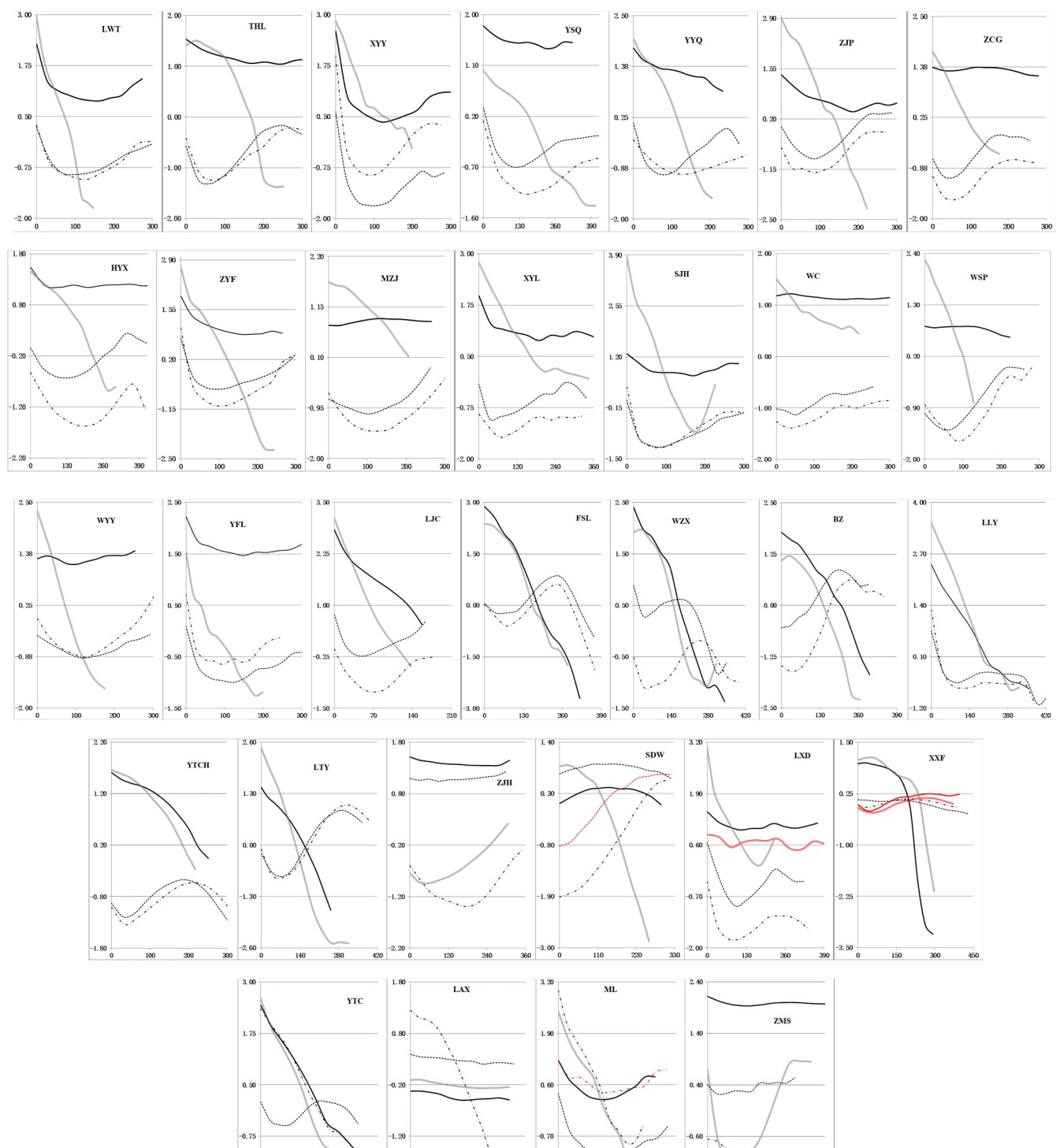

(Note: Single line stands for T1a, double line for T1b, dotted line for T2, pecked line for T3, red line for the correspondeing tone category variation.)

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
