# Peer review of "Phonetic Tonal Manifestations and Trends in Tone Change: A Case Study of the Yong-Deng Dialect in Northwest China"

_languages, doi:10.3390/languages8040262_

Round 1

Reviewer 1 Report

Comments and Suggestions for Authors

This manuscript reports tonal production data from two dialect points of Northwest Mandarin Chinese. The results provide novel data to the literature on acoustical tonal manifestation and trend of tone change and the discussion is also informative. However, to bring the manuscript up to the level of publication, I would suggest revisions regarding the following issues:

1. The title of this manuscript specifically indicates that this study worked on the acoustic tonal manifestation of tone change, but there is no sign of any acoustical analyses reported in this paper. The author(s) should at least mention what types of acoustical analyses they had conducted and how the acoustical data had supported their claims on the tone changing patterns. Did the author(s) just observe the F0 contours without any further analyses?

2. It is confusing when the author(s) put Section 3.1 in the results, because the clarification of concepts should appear in the introduction.

3. Data from two dialect points were collected, but there was a huge difference in the number of speakers for each point. The author(s) should pay attention to this as the HY data might be biased due to the small number of speakers.

4. The author(s) mentioned the three-syllable combination in Line 131, but it was clearly shown from the materials that only monosyllabic and disyllabic words were included. So where did the three-syllable combination come from?

5. In Line 187, it is not well articulated how the author(s) deduced the proto tone features from their results.

6. It is confusing when the author(s) used both “=” and “>” in the manuscript, although some explanations were provided in Lines 199 to 202.

7. In Lines 291 – 293, more justifications should be provided as to the author(s)’ claim that “the directionality of tonal change should have less to do with ‘clock-wise’ or ‘counter clock-wise’ circle than with the phonological rules governing the tonal change”.

8. The language and formatting of this manuscript should be improved significantly. The sentences are in general very difficult to follow and there are grammatical errors that should have been corrected before submission (e.g., “One thing need to clarify” in Line 230 should be “One thing needs to be clarified”; the use of the casual “By the way” in Line 180). The formatting is not consistent throughout this paper (e.g., the caption of Table 7; Line 350).

9. Some cited papers (e.g., Zhu and Yi (2015) on Line 46; Lin (1995) in Table 5) are not listed in the reference list. Also, the authors should cite the Fangyan diaocha zibiao in the reference (Line 80).

Author Response

Dear reviewer, 

I have made revisions according to your comments and your suggestions. Please see the 'response letter' in attachment. 

Reviewer 2 Report

Comments and Suggestions for Authors

Review of: Acoustic tonal manifestations and trend of tone change: a case-study of Hai-yuan and Yong-deng

Summary & Background Assumptions:

The author sets out to study the patterns of tonal mergers in Northwest Chinese dialects.   To this end, the Hai-yuan (HY) and Yong-deng (YD) dialects serve as a case study. The author examines the monosyllabic tonal patterns of these dialects, integrates evidence drawn from literary sources, and compares these dialects to neighboring varieties.  Special attention is given to the tone sandhi rules in each variety.  The author’s method is informed by recent advances in our understanding of the likely directionality of diachronic patterns of tonal change (See Hyman 2017 - Tones spread from left to right, contour tones tend to level out, Low-High sequences tend to compress, and High-Low sequences generally expand).  The author’s assumptions are further supported by work by Pittayaporn (2018) who argues that patterns of synchronic tonal variation gradually become phonologized, and thus drive diachronic change.  The author also cites the work of Yang and Xu (2019) who have advanced a diachronic typology of tonal change (clock-wise route) that is based on the most common tonal changes that are observed in many languages. In the end, however, the author concludes that the tonal changes observed in HY and YD (especially tone contour changes) fit more closely with the tone-evolution clock model proposed by Zhu (2018). 

Data:

Data on the pronunciation of tone in monosyllabic, disyllabic, and word tone in both dialects were gathered by the author (10 speakers of HY and 31 speakers of YD).  

Monosyllabic vs. Disyllabic tonal patterns: (Section 3.2.1)

The author lists the current disyllabic tonal patterns of both the HY and YD varieties (section 3.2.2) and then posits a reconstruction of the original tonal pattern (which the author calls the original “prototype”) of each category.  These reconstructions were based on a comparison of 1) written records of the HY and YD varieties and by 2) comparing the tonal patterns of neighboring dialects and, finally, by 3) observing the tonal patterns of these dialects when alternating between monosyllabic and disyllabic forms.  The author lists the resulting reconstructions of tones in monosyllables in his/her Table 5.

(See Table 5 in author's text.)

Methodology:

I find that the methodology used in this paper was appropriate.  Data on the tones used in HY and YD were gathered from native speakers of these varieties and the amount of data was likely sufficient.  The proto-tones in question were reconstructed in an appropriate manner using both synchronic data and philological sources.  The author’s comparison of the synchronic alternations of tones in monosyllables and disyllables was indeed advisable.  The author sought to apply different theories of tonal change to his study.  However, he found that the diachronic changes in tone he discovered were in line with the theory advanced by Zhu (2018).

Limitations of this review:

The questions posed by the author in this paper are certainly interesting is light of the current state of research into the directionality of diachronic tonal changes.  I am unfortunately limited in my ability to assess the present paper, however, because I do not have extensive expertise in Chinese dialectology nor in the diachronic development of Sinitic languages.  I am thus not able to independently verify some of the key assumptions that the author makes in this paper such as the choice of selecting HY and YD for use in this study (as opposed to other varieties).  Obviously, I also cannot vouch for the accuracy of the data that were collected.

Recommendation

I recommend publication of the paper assuming that the editors are comfortable with the accuracy of the data collected and the method of tonal reconstruction used by the author.

Literature:

Hyman, Larry M. (2019) Synchronic vs. diachronic naturalness: Hyman and Schuh (1974) revisted. UC Berkeley Phonetics and Phonology lab report (2017).In Margit Bowler, Philip T. Duncan, Travis Major & Harold Torrence (eds.), Schuhschrift: Papers in Honor of Russell Schuh, 50-65. eScholarship Publishing, University of California. (https:scholarship.org/uc/item/7c42d7th

Pittayaporn, Pittayawat. (2018). Phonetic and systemic biases in tonal contour changes in Bangkok Thai. In H. Kubozono & M. Giriko (Eds.) Tonal Change and Neutralization (pp. 249-278). Berlin: Mouton de Gruyter.

Yang, Cathel and Xu, Yi. (2019). Cross-linguistic trends in tone change: a review if tone change studies in East and Southeast Asia. Diachronica 36: 417-459.

Zhu, Xiaonong. (2018). The evolutionary comparative method: How to conduct studies of tonal evolution?  Linguistic Science 2:113-132.

Round 2

Reviewer 1 Report

Comments and Suggestions for Authors

The author has addressed the issues raised by the reviewers and the quality of this manuscript has been greatly improved. However, there are some minor issues for the authors to consider:

1) In Lines 77-79, the author mentioned that there are different monosyllabic tonal patterns in YD Mandarin, which is also shown in Table 1 (Line 126). A closer observation from Table 1 suggests that there was only one representative speaker for Types 4 to 7, and the background information of these four speakers varies a lot. I wonder if there is any literature that documents the seven tonal patterns in YD Mandarin. If not, the probability of 1/31 may not be strong enough to represent and argue for a separate tonal pattern.

2) The author should provide more details in the Materials and Methods section. It would be good to separate this section into subsections, e.g., participants, materials, procedures, data analysis, etc. In addition, it is unclear how the tonal categories were identified. The design of the Perception test and correlation analysis should also be included in this section.

3) The correlation analysis reported in 3.2.2. should be replaced by other statistical analyses because the variables (gender, education and tonal category) are not numerical. They should be regarded as categorical data.

4) For the discussion of the data, the author may also consider the influence of language contact, particularly the dominance of Standard Mandarin in China, which may have affected the tonal realisation of YD Mandarin by different speakers.

5) Although the language has been improved, it is suggested that the author send out the manuscript for proofreading, especially the added contents. For example, in Lines 61-62, “tonal change study” should be “studies on tonal change”.

Author Response

  • In Lines 77-79, the author mentioned that there are different monosyllabic tonal patterns in YD Mandarin, which is also shown in Table 1 (Line 126). A closer observation from Table 1 suggests that there was only one representative speaker for Types 4 to 7, and the background information of these four speakers varies a lot. I wonder if there is any literature that documents the seven tonal patterns in YD Mandarin. If not, the probability of 1/31 may not be strong enough to represent and argue for a separate tonal pattern.

Great thanks to the reviewer's comments. The author has revised this part. The tone pattern identified 17 speakers was defined as the basic type of YD dialect, and the other six different tone patterns were defined as its variants. YD dialect was recorded in the literature a 4-tone system in most cases. It is different from our survey, which is also one of the reasons that we wrote this article. In addition, the author does agree with the reviewer's opinion that the probability of 1/31 is certainly not sufficient to represent the tone pattern of a dialect. While for these individuals, the term ‘tone pattern’ used in this article only refers to their personal tonal system.

  • The author should provide more details in the Materials and Methods section. It would be good to separate this section into subsections, e.g., participants, materials, procedures, data analysis, etc. In addition, it is unclear how the tonal categories were identified. The design of the Perception test and correlation analysis should also be included in this section.

Great thanks to the reviewers' advices. The author has rearranged the section "Materials and Methods". Besides the introduction of participants, materials, procedures and data analysis, the author also clarify how the tone categories in Chinese dialects were defined and distinguished, as well as the general characteristic of tone categories in Northern mandarin dialects. The design of the Perception test and correlation test is also put in this section.

  • The correlation analysis reported in 3.2.2. should be replaced by other statistical analyses because the variables (gender, education and tonal category) are not numerical. They should be regarded as categorical data.

Thanks to the reviewer's comments. The author replaced the pairwise correlation analysis to a crosstab correlation analysis, and provided a frequency contingency table between variables and tone-pattern choices. To further confirm the degree of correlation, three chi-square tests were performed as well.

  • For the discussion of the data, the author may also consider the influence of language contact, particularly the dominance of Standard Mandarin in China, which may have affected the tonal realisation of YD Mandarin by different speakers.

Thanks to the reviewer’s comments. The author adds this part to the conclusion:

Yi (2019) found that in Lanzhou dialect, with the two variations of T1a (high level and high falling), people under 50 years old and people with higher education are more inclined to adopt the level variant. It is within the late 50 years that Standard Mandarin (Putonghua) began to take its prestigious position. In YD, a parallel transformation can also be observed in T1a. Among 31 speakers, 23 opt for the high level variant. And it retains a high level in almost all positions of the disyllabic and tri-syllabic combinations. In perception test, of all the choices, 50% took high level [H] as the ‘accepted’ tone feature and only 36% chose high falling. It seems that T1a does have been influenced by Standard Mandarin, experiencing the alternation from a high-mid falling to a high level. However, the other three tonal categories in YD showed little evidences of the same influence. Besides, the tonal change from a high falling to a high level also occurs in other dialects/languages and also in other tonal categories (Yang & Xu 2019), where there is not the influence from Chinese Mandarin. I admit that language contact does play a certain role in tone change, while, based on these observations, I propose that the directionality of tone change is influenced more by the phonological rules than by adhering strictly to a 'clock-wise' or 'counter-clockwise' directionality or by the influence from certain language contact.

  • Although the language has been improved, it is suggested that the author send out the manuscript for proofreading, especially the added contents. For example, in Lines 61-62, “tonal change study” should be “studies on tonal change”.

Thanks to the reviewers' comments. The author has sent out the article and had it proofread by a professional native speaker. The errors have been corrected.
